# Depression Accompanied by Hopelessness Is Associated with More Negative Future Thinking

**DOI:** 10.3390/healthcare12121208

**Published:** 2024-06-17

**Authors:** Hailong Han, Akira Midorikawa

**Affiliations:** 1Department of Psychology, Graduate School of Letters, Chuo University, Tokyo 192-0393, Japan; 2Department of Psychology, Chuo University, Tokyo 192-0393, Japan; green@tamacc.chuo-u.ac.jp

**Keywords:** episodic future thinking, depression, hopelessness, hopelessness depression, future thinking task

## Abstract

Background: This study aimed to delineate the interplay between depression, hopelessness, and episodic future thinking (EFT), focusing on cognitive biases towards negative future thinking that are central to depressive symptomatically. Methods: A Japanese university student was utilized to scrutinize divergences in EFT across groups stratified by varying degrees of depression and hopelessness. The research leveraged a modified future thinking task (FTT), the Beck Hopelessness Scale, and the Beck Depression Inventory-II to gauge participants’ levels of hopelessness and depressive symptoms. Results: Consistent with prior research, the non-depressed group showed a reduction in positive EFT, reinforcing the idea that diminished positive future thinking is a hallmark of depressive conditions, even in the absence of a clinical diagnosis. Moreover, individuals with comorbid depression and elevated hopelessness demonstrated a significant decrease in positive EFT and an increase in negative EFT, substantiating a distinctive cognitive profile for this subgroup. This finding suggests that the presence of hopelessness exacerbates the negative cognitive biases associated with depression. Conclusions: The study emphasizes the importance of considering hopelessness as an independent construct when assessing EFT in clinical contexts. The pronounced impact of hopelessness on future thinking in those with depression suggests that targeted interventions, such as future-directed therapy (FDT), may be particularly effective for individuals with hopelessness depression by focusing on modifying negative future thinking patterns and enhancing life quality.

## 1. Introduction

Pessimistic future thinking is a core component of depression [1,2,3]. Therefore, exploring future-related cognitions may improve our understanding of depression. Episodic future thinking (EFT) refers to the ability to mentally project oneself into the future to pre-experience future events [4]. Many clinical and non-clinical studies have been conducted on the relationship between EFT and depression [5,6,7,8,9].

MacLeod et al. devised a future thinking task (FTT) based on the verbal fluency paradigm to explore EFT [10]. The FTT involves three time periods: 1 week, 1 year, and 5–10 years. Participants were asked to describe positive and negative events that they thought might occur in the future, within 1 min for each time period. No difference in negative EFT was observed between depressed patients and controls. However, a decrease in positive EFT was seen in depressed patients compared with non-depressed people. Similar results have been reported by other researchers [11,12,13,14], including in studies using paradigms other than the FTT [15,16,17]. In summary, a less positive view of the future is considered a core characteristic of depression [18]. 

The belief that one does not have a positive future is referred to as hopelessness. In some studies where hopelessness was considered independently, it was found that respondents with higher hopelessness scores also showed a lack of positive future thinking and no corresponding increase in negative future thinking [16,19,20]. MacLeod et al. reported a stronger correlation between an increase in hopelessness and a decrease in positive EFT compared to negative EFT [9]. In addition, the results of the correlation analysis showed that positive EFT was more strongly correlated with hopelessness than positive EFT and depression. Based on this correlation difference finding, the decrease in positive EFT may stem from the combined effects of hopelessness and depression. However, we have found a question that is worth considering. A link between depression and hopelessness has frequently been reported [21], but they do not always appear simultaneously. The DSM-IV and ICD-10 diagnostic criteria for depression include depressed mood, loss of interest or pleasure in almost all activities, and other symptoms. However, the DSM-5 has expanded the core mood criteria to include hopelessness [22,23]. Mood symptoms are not the only necessary condition in the diagnostic criteria. Therefore, although hopelessness is a common symptom of depression, not all people with depression experience hopelessness, and not all people who experience hopelessness suffer from depression [24]. Based on hopelessness theory, Abramson et al. argued that depressed patients with a high degree of hopelessness should be separated from the concept of depression, and proposed the concept of hopelessness depression [2]. They considered hopelessness depression as a subtype of depression and argued that individuals with hopelessness depression have an independent cause, symptoms, course, and treatment. Although hopelessness and depression are strongly correlated, they should be considered two relatively independent dimensions. When combining these dimensions, four classifications emerge, as shown in Figure 1 (Low-Hopelessness Depression, LH-D; Hopelessness Depression, HD; Low-Hopelessness Non-Depression, LH-ND; Hopelessness Non-Depression, H-ND). If we consider the effects of hopelessness and depression on EFT from such a perspective, there may also be relatively independent components. However, some studies exploring the effects of depression on situated future thinking have treated hopelessness as a necessary symptom of depression, ignoring the possibility that there are differences in feelings of hopelessness in groups. We argue that such overlooked differences may have an impact on specific outcomes. 

This paper focuses on two central questions to make the case. First, we will confirm whether the results are consistent with prior studies if depression is used as a screening condition for participants; that is, whether depressed groups are characterized by reduced positive EFT compared to non-depressed groups. In addition, as a cross-cultural comparison, this study was conducted in Japan and with Japanese subjects. The North American study used the same paradigm to examine whether the same reduction in positive EFT is present in the characteristics of depressed patients under East Asian cultural conditions. Second, we aim to examine potential differences in positive EFT or negative EFT between the HD and LH-D groups under the consideration that depression and hopelessness are distinct dimensions. The constructive Episodic Simulation Hypothesis argues that episodic future thinking relies on the episodic memory system. Imagining future events is a constructive process of piecing together fragments of memory collected from various experiences and events (objects, people, actions, places, emotions) [25]; that is, when hopelessness has a specific effect on the emotional valence of the content of memory extraction, this effect is the same for future thinking. Some studies have found that feelings of hopelessness affect memory recall. Increased feelings of hopelessness are associated with decreased retrieval of positive memories and increased retrieval of negative memories [26,27]. Therefore, the HD group may be accompanied by the possibility of increased negative EFT along with decreased positive EFT.

## 2. Method

### 2.1. Participants

Participants were recruited from among students enrolled in a psychology course from various faculties of Chuo University, Japan. Initially, 71 subjects participated in the experiment. However, five subjects were excluded from the final analysis due to the following reasons: inability to concentrate on the experiment, interruptions during the experiment, inability to continue due to headaches, and inability to continue due to mental distress caused by negative future thinking. Ultimately, 66 subjects (32 males and 34 females) with a mean age of 19.95 ± 1.28 years (range: 18–23 years) were included in the final analysis.

### 2.2. Materials

Future thinking task. The FTT was adapted from the verbal fluency paradigm of MacLeod et al. [9], and translated into Japanese by us. Participants were asked to think about and verbalize future events that they thought would occur during three time periods: the next week, the next year, and 5–10 years’ time. There were positive and negative future experience conditions. Participants were requested to generate as many EFT events as possible within 1 min for each of the three time periods under both conditions. 

Verbal fluency task. The verbal fluency task (VFT) served as a control for the FTT, and the procedure was based on Saito et al. [28]. The VFT comprises a category fluency task (CFT) and a letter fluency task (LFT). The CFT used three categories (animal, fruit, and vehicle). For the LFT, the participants were asked to generate words beginning with “あ[a]”, “か[ka]”, and “し[shi]” [29], excluding the names of people or places (proper nouns) and numbers. Duplicates were also discounted. Each trial lasted 1 min and participants were instructed orally by the experimenter to generate words.

Beck Hopelessness Scale. Participants were asked to complete the Japanese version of the Beck Hopelessness Scale (BHS) [30]. The BHS is a 20-item self-report scale developed by Beck et al. [31] to assess the level of despair about the future. Participants were asked to respond on a 2-point (true or false) scale, and the maximum score was 20 points.

Beck Depression Inventory-II. The Japanese version of the Beck Depression Inventory-II (BDI-II) [32] was used to evaluate depression (adapted from the English version of the BDI-II) [33]. The BDI-II was developed as a self-report measure of the presence and severity of depressive symptoms according to the DSM-IV diagnostic criteria; it consists of 21 items rated on a 4-point scale (range: 0–3) based on the last 2 weeks. Scores < 14 are considered normal.

### 2.3. Procedure

The experiment was conducted individually with each participant between October 2019 and January 2020 and lasted approximately 60 min. The experiment started with the FTT, and the order of the positive and negative conditions was randomized. Before beginning the experiment, the following message was relayed to the participants: “If you feel uncomfortable or distressed by negative future thoughts, you can stop the experiment at any time”. The VFT was also performed, and the scores for the CFT and LFT components were recorded. The participants then completed the BDI-II, followed by the BHS and debriefing.

### 2.4. Groups

Two different groupings were required to test each of the two hypotheses separately. The first grouping method utilized BDI-II scores, with a cutoff score of 14, to categorize the 66 participants into two groups: depressed and non-depressed. Specifically, the depressed group comprised 29 participants, while the non-depressed group included 37 participants. Second, the 66 participants were divided into four groups based on the BDI-II cutoff score and mean BHS score reported for the young Japanese population (8 points) [34]. The groups were as follows: Hopelessness Depression group (HD; *n* = 14), Low-Hopelessness Depression group (LH-D *n* = 15), Hopelessness Non-Depression group (H-ND; *n* = 20), and Low-Hopelessness Non-Depression group (LH-ND; *n* = 17).

## 3. Results

### 3.1. Demographic and Self-Report Scale Data

The mean and standard deviation demographic and self-reported scale data for each group are shown in Table 1. One-way analysis of variance (ANOVA) was used to compare the groups in terms of age, verbal fluency, and the BDI-II and BHS scores. The chi-square test was used to test for gender differences. Age and VFT scores were not significantly different among the groups. ANOVA confirmed intentional differences in BDI-II scores (F[3, 62] = 45.5, *p* < 0.001, η*p*^2^ = 0.69). The results of multiple comparisons showed that intentional differences were found in the results of comparisons between groups, except between HD and LH-D, where no intentional differences were found. In addition, intentional differences were confirmed in BHS scores (F[3, 62] = 78.4, *p* < 0.001, η*p*^2^ = 0.79). The results of the multiple comparisons showed that intentional differences were found in the results of the comparisons between groups, except between H-ND and LH-D, where no intentional differences were found. Finally, no intentional differences were found for gender differences either (*p* = 0.90).

### 3.2. Group Differences in EFT between Depression and Non-Depression

To examine whether depressed participants would experience a reduction in positive EFT when depression was used as a screening condition. A 2 (group: depressed/non-depressed) × 2 (Valence: positive/negative) ANOVA was used to compare the number of FTT events produced by the two groups in the positive and negative conditions. Before conducting the ANOVA, we checked for normality and homogeneity of variances to ensure the data met the basic assumptions required for ANOVA. Normality was checked using the Shapiro-Wilk test, and homogeneity of variances was assessed using Levene’s test. The normality test results showed that for the Positive Affect (PA) group, the Shapiro-Wilk statistic was 0.97 (*p* = 0.06), and for the Negative Affect (NA) group, it was 0.97 (*p* = 0.07). Although the *p*-values are close to the significance level, they are greater than 0.05. Therefore, we do not reject the null hypothesis of normal distribution. It seems that even though the sample sizes were small, the data in both groups seems to follow a normal distribution. Levene’s test for equality of variances indicated that there were no significant differences in variances between the non-depression and depression groups, with *p*-values of 0.98 and 0.29 for the PA and NA groups, respectively. Valence was a within-group factor, and the results are shown in Figure 2. There was a main effect of valence (F[1, 64] = 14.6, *p* < 0.001, η*p*^2^ = 0.19): the number of positive EFT events exceeded the number of negative ones. A group × valence interactive effect (F[1, 64] = 18.1, *p* < 0.001, η*p*^2^ = 0.22) was also seen. The results of the multiple comparisons (Holm correction) revealed that, although the two groups did not differ in the number of negative EFT events, there was a significant decrease in the number of positive EFT events in the depression group compared to the non-depression group.

### 3.3. Group Differences in EFT between the Four Groups

A 4 (group: HD/LH-D/H-ND/LH-ND) × 2 (valence: positive/negative) ANOVA was used to test for differences in EFT between the four groups. Before conducting the ANOVA, we checked for normality and homogeneity of variances. The Shapiro–Wilk test showed that the Positive Affect (PA) group had a value of 0.97 (*p* = 0.06), and the Negative Affect (NA) group had a value of 0.97 (*p* = 0.07). Both *p*-values were higher than the conventional alpha level of 0.05, indicating no violation of the assumption of normality. Additionally, Levene’s test for equality of variances indicated no significant differences across the groups, with *p*-values of 0.98 for the PA group and 0.79 for the NA group. This confirmed that the variances were equal across the different levels of hopelessness and depression. Valence was a within-group factor, and the results are shown in Figure 3. A main effect of potency was still found (*F*[1, 62] = 9.66, *p* < 0.005, η*p*^2^ = 0.14), and the results of the multiple analyses indicated that the total number of positive EFT events was significantly greater than the number of negative EFT events. A group × valence interaction was also found (*F*[3, 62] = 10.23, *p* < 0.001, η*p*^2^ = 0.33), with significant differences between groups for both positive EFT (*F*[3, 124] = 6.14, *p* < 0.001, η*p*^2^ = 0.23) and negative EFT (*F*[3, 124] = 4.23, *p* < 0.005, η*p*^2^ = 0.17). Significant differences in positive EFT were found by multiple comparisons between both depression and non-depression groups, with the H-ND group having a significantly higher number of positive EFTs than the HD (*p* < 0.01) and LH-D (*p* < 0.05) groups. The LH-ND group was also significantly higher than the HD (*p* < 0.05) and LH-D (*p* < 0.05) groups. On the other hand, the results of multiple comparisons of negative EFT showed that the HD group had significantly higher numbers than the H-ND (*p* < 0.05), LH-ND (*p* < 0.05), and LH-D (*p* < 0.05) groups. There were no other significant differences found.

Additionally, the Holm correction was applied during the multiple comparisons stage. For pairwise comparisons with significant differences, the unadjusted and adjusted *p*-values are shown in Table 2.

## 4. Discussion

This study explored whether there are differences in positive and negative EFT between depression with and without hopelessness, hopelessness without depression, and control groups. We hypothesized that there would be a difference in EFT between the depression with and without hopelessness groups; specifically, that the depression with hopelessness group would have less positive EFT than the depression without hopelessness group; and that EFT in the non-depression with hopelessness group would be significantly less positive compared to the control group.

We observed a significant difference in future thinking between the D/HH and D/LH groups: the number of positive EFT thoughts was significantly lower in the former group. Second, consistent with previous studies, non-depressed participants (ND/HH and ND/LH groups) produced more positive responses overall. Third, there was a trend toward a decrease in positive EFT in the D/LH group compared to the non-depressed groups (ND/HH and ND/LH), but not in the negative EFT responses. These results replicate previous studies [3,10,11,12,14] and provide further support for the view that depression is characterized by cognitive biases with respect to future events [35]. In addition, the subjects with depression and high hopelessness exhibited more negative EFT than those in the other groups. Therefore, we suggest that depression with high hopelessness is cognitively dissociable from depression with low hopelessness. The depression with high hopelessness group exhibited both a decrease in positive EFT and an increase in negative EFT.

Abramson et al. concluded that hopelessness depression consists of delayed initiation of autonomous responses (motivational symptoms) and sadness (emotional symptoms) [2] and they argued that less positive expectations were the main cause of the motivational symptoms. In contrast, sadness stems from negative expectations for the future. The current findings support Abramson et al. with respect to the symptoms of hopelessness depression. Furthermore, Iacoviello et al. highlighted that hopelessness, self-blame, brooding/worry, and decreased self-esteem are the earliest symptoms to manifest in HD episodes and the latest to remit [36]. The current study strengthens their argument that hopelessness and associated cognitive symptoms are central to HD.

### Limitation and Future Research

Several limitations of the present study should be discussed. First, the effect of anxiety was not considered in the context of the FTT paradigm. MacLeod and Byrne noted significantly more negative EFT in anxious participants than in a control group [3]. Therefore, we could not confirm whether the increase in negative EFT seen in our D/HH group was attributable to anxiety, where anxiety and depression are frequently comorbid. The effect of anxiety should be assessed in future studies. Second, suicide, which is a major symptom of hopelessness depression [2,37], was not considered in this study. Future studies should include participants with despairing depression.

Hopelessness depression is associated with a higher risk of suicide compared to other types of depression, such that exploring EFT in this group was valuable. We found that EFT differed according to the presence of hopelessness and/or depression; as such, hopelessness should be considered in future studies. Based on the lack of positive EFT in depression, an interventional approach, future-directed therapy (FDT), has been proposed [38,39]. FDT aims to reduce depression symptoms and improve well-being by changing thinking patterns, reducing dwelling on the past or focusing on inappropriate goals, and improving expectations for the future. However, whether FDT can reduce negative future thinking and its effectiveness for treating hopelessness depression remain to be determined. Interventions and treatments for hopelessness depression need to be further refined.

## 5. Conclusions

The present study has provided substantial insights into the complex interplay between depression, hopelessness, and episodic future thinking (EFT). Our findings confirm that individuals suffering from depression, particularly when compounded by hopelessness, exhibit a pronounced tendency towards negative future thinking. This aligns with existing literature that identifies impaired positive future thinking as a characteristic cognitive bias in depression. Importantly, our research extends these insights by distinguishing the effects of hopelessness within depressive states. Participants with comorbid depression and hopelessness demonstrated not only reduced positive EFT but also an enhanced inclination towards negative anticipations about the future. This distinction is crucial, as it suggests that hopelessness and depression, while often co-occurring, influence cognitive processes related to future thinking in uniquely detrimental ways.

This study’s emphasis on the dual dimensionality of depression and hopelessness underscores the necessity for clinical approaches that specifically address the nuanced cognitive profiles of these conditions. Interventions like future-directed therapy (FDT) could be particularly beneficial, focusing on modifying negative future thinking patterns and fostering more positive future expectations. This approach may enhance the overall life quality and coping strategies of individuals affected by hopelessness depression. Furthermore, the cross-cultural aspect of this research, focusing on a Japanese cohort, contributes to a more global understanding of these psychological constructs, which often are studied within Western contexts. It underscores the need for culturally sensitive diagnostic and treatment strategies that consider the specific sociocultural dynamics affecting mental health.

In conclusion, while the intertwined effects of hopelessness and depression on EFT complicate the clinical picture, they also open avenues for targeted, nuanced therapeutic interventions. Future research should continue to explore these dynamics across diverse populations and develop interventions that not only alleviate symptoms but also empower individuals to envision and strive for a hopeful future.

## Figures and Tables

**Figure 1 healthcare-12-01208-f001:**
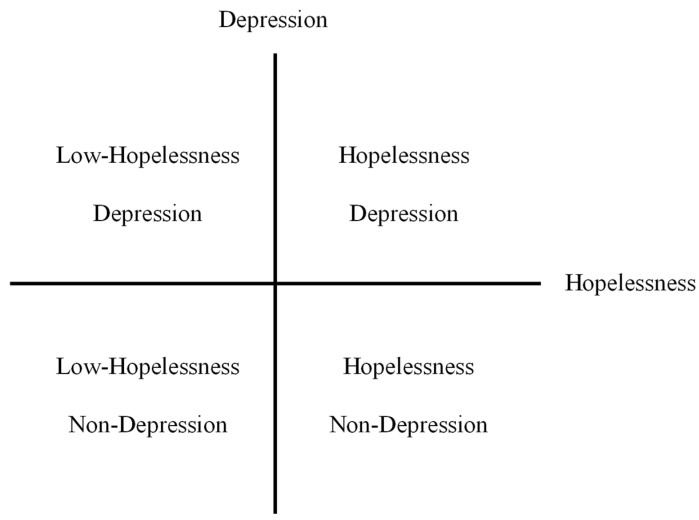
Four classifications of depression and hopelessness combinations.

**Figure 2 healthcare-12-01208-f002:**
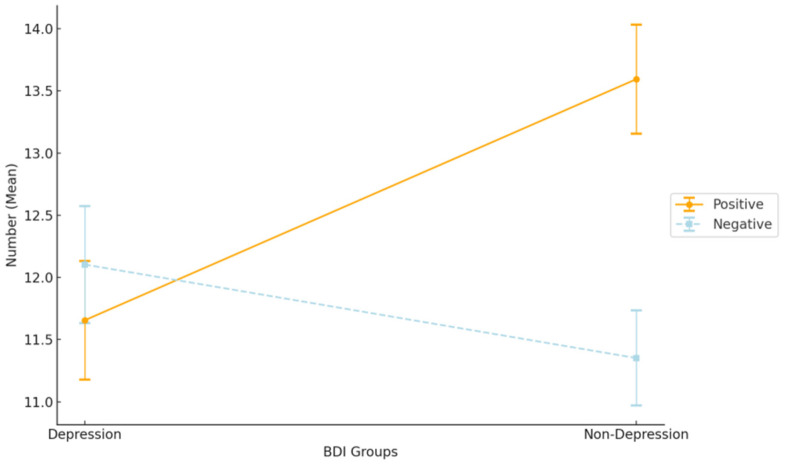
Mean numbers of positive and negative EFT events between the depression and non-depression groups.

**Figure 3 healthcare-12-01208-f003:**
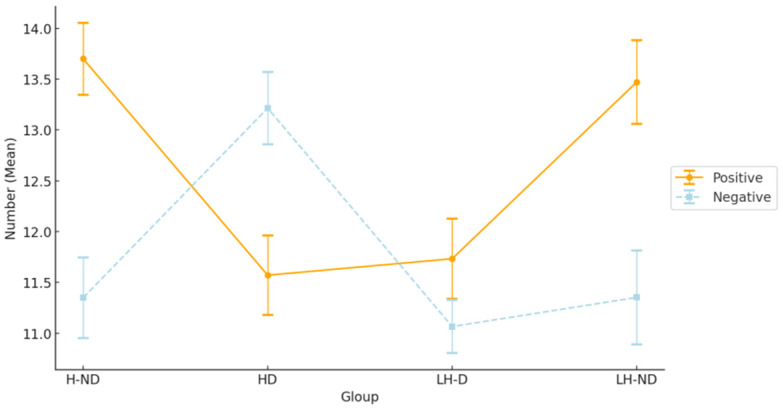
Mean numbers of positive and negative EFT events in each group.

**Table 1 healthcare-12-01208-t001:** Demographic and clinical data of the groups.

Variable	HD (*n* = 14)	LH-D (*n* = 15)	H-ND (*n* = 20)	LH-ND (*n* = 17)
Age (y), mean (SD)	19.7 (0.8)	20.1 (1.2)	19.8 (1.4)	20.2 (1.6)
range	18–23	19–21	18–23	19–23
Gender, n (%) male	8 (57.1)	6 (40.0)	9 (45.0)	9 (53.0)
Verbal fluency total score, mean (SD)	70.9 (8.6)	66.4 (7.8)	64.6 (7.6)	63.8 (7.7)
BDI-II score, mean (SD)	19.5 (4.4)	17.1 (2.8)	10.0 (3.1)	7.4 (3.1)
BHS score, mean (SD)	11.8 (1.7)	7.6 (1.4)	8.2 (0.9)	5.4 (0.7)

Note: HD: Hopelessness Depression; LH-D: Low-Hopelessness Depression; H-ND: Hopelessness Non-Depression; LH-ND: Low-Hopelessness Non-Depression.

**Table 2 healthcare-12-01208-t002:** Multiple Comparisons Results with Holm Adjustment.

Comparison	Adjusted *p*-Value
Positive	
H-ND vs. HD	0.007
H-ND vs. LH-D	0.011
LH-ND vs. HD	0.019
LH-ND vs. LH-D	0.025
Negative	
H-ND vs. HD	0.021
LH-ND vs. HD	0.023
HD vs. LH-D	0.012

Note: The table shows Holm-adjusted *p*-values for each hypothesis test.

## Data Availability

The data presented in this study are available on request from the corresponding author due to privacy.

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
