# Peer review of "Depression Accompanied by Hopelessness Is Associated with More Negative Future Thinking"

_healthcare, 2024, doi:10.3390/healthcare12121208_

Round 1
Reviewer 1 Report
Comments and Suggestions for Authors
1. How was it determined that the discarded participants were not focusing on the experiment?
2. How was sample size determined? There needs to be an a-priori power analysis, and then an a-posteriori power analysis to determine what the actual power was, and how much was the probability of a type-II error
3. What was Cronbach's alpha for Beck's Hopelessness and Beck's Depression scale?
4. In mentioning the Holm correction, there should be a specification what the new alpha level is aftre the correction.
5. In order to make the interaction effect (or lack thereof) more clear, I suggest switching Figures 2 and 3 to a type of graph where the lines either run parallel or intersect. That makes visualization of testing for interaction much clearer.
6. The limitations should be discussed in a separate section.
7. In presenting the ANOVA analyses, there should be a discussion of whether the tests met the necessary assumptions (especially normality and homogeneity of variance, given the small sample size).
Author Response
Thank you for your insightful comments and we have made new adjustments to the manuscript based on your questions. However, the detailed answers to your questions have been organized in a Word document and added as an attachment. Finally, thank you again for your comments and feedback.

Reviewer 2 Report
Comments and Suggestions for Authors
Thank You for the opportunity of reviewing the paper. The issue of depression and factors related to it are very important for the perspective of modern society.
The group of participants is relatively small (66 people). What was the critieria for invitation those people?
They were a students of what faculty/ies? Could IT affect the results?
What about their previous experiences? Did they had some MH problems before?
Some technical issues, like (Leod et al) wrote in different font. I suggest to check the paper from this point of view.
What about Japanaise version of FFT? If it's not exists, what version has the Authors used?
How many participants were in each of the groups (A1-A4)?
Based on the statistical analysis-, the groups had normal disturbence?
Discusion- the literature that the Authors support their findings is quite old (1989, the newest was published in 2013).
Author Response

(The authors gave the same response as above.)

Round 2
Reviewer 2 Report
Comments and Suggestions for Authors
Thank You for the changes that the Authors included in their paper.